# Determining the Effectiveness of *Saccharomyces cerevisiae* as a Postbiotic in Mass-Reared *Acheta domesticus* (House Cricket)

**DOI:** 10.3390/insects16070702

**Published:** 2025-07-09

**Authors:** Kimberly L. Boykin, Erik Neff, Mark A. Mitchell

**Affiliations:** 1Department of Veterinary Clinical Sciences, Louisiana State University, Skip Bertman Dr, Baton Rouge, LA 70803, USA; mmitchell@lsu.edu; 2Fluker’s Cricket Farm, Inc., 1333 Plantation Avenue, Port Allen, LA 70767, USA

**Keywords:** feed additive, commercial insect farming, total biomass, cricket viruses, 16S/18S microbiome

## Abstract

Nutritional supplements such as pre-, pro-, and post-biotics have been used in several feeder insect species to increase yields and improve health outcomes. This study provided commercially reared crickets with a *Saccharomyces cerevisiae* postbiotic product and found that a 0.5% inclusion rate within their normal diet was able to increase survival and produce a higher total biomass. No changes were seen in viral prevalence rates, but there were differences found within the gut microbiome that likely contributed to increased survival rates. Overall, the inclusion of a postbiotic appeared to be beneficial and further study is warranted.

## 1. Introduction

Insects have been considered a staple food item in some cultures for thousands of years but have only recently started gaining popularity in the Western world [1]. Their popularity is mainly due to their sustainable nature as a source of animal protein for both humans and animals [1,2,3]. Unfortunately, the feeder insect industry has been plagued by disease outbreaks that have impacted the supply of insects and thus limited their ability to be used for human and animal consumption. Over the last several decades, *Acheta domesticus* commercial production has been severely impacted by viral diseases with some epizootics resulting in 100% mortality rates [4,5,6]. *A. domesticus* densovirus (AdDV), *A. domesticus* volvovirus (AdVVV), and invertebrate iridovirus 6 (CrIV) are known to be highly prevalent on commercial cricket farms and AdDV and CrIV are known to be associated with high morbidity and mortality rates [4,5,6,7,8]. Similarly to other agricultural markets, high animal densities and management practices put insects at high risk for infectious diseases [3]. If the insect industry is going to be viable and provide animal protein to growing populations of humans and animals, it is vital that we develop a better understanding of the epidemiology of insect diseases and how best to improve their health.

There is a growing trend among researchers from several different disciplines to measure the importance of the gut microbiome. In humans and animals, a healthy and balanced microbiome is essential in supporting the overall health of the individual. There is strong evidence from several insect species that microbial communities are not only beneficial in digestive processes, but that they perform essential functions for host physiology related to immunity, regulation of metabolism, and even neurobehavioral traits [9,10,11,12,13]. When animals experience a disruption to their normal microbial community structure (intestinal dysbiosis), animal health and fitness are often affected [10,13]. Pre-, pro-, and postbiotics are often used as feed additives to improve the health of the gut microbiome in humans and animals, including insects [13,14]. Multiple studies have already shown their safety and efficacy in traditional livestock species, as well as several species of feeder insects. Increased larval growth, enhanced reproduction, and positive immunomodulating effects are common outcomes for insects that have been provided probiotics during their development [13]. *Saccharomyces cerevisiae*, a commonly used fungal probiotic, has been used successfully in silkworms (*Bombyx mori*) to increase silk yield, in black soldier fly larvae (*Hermetia illucens*) to increase bioconversion rates and improve nutritional profiles, and in yellow mealworms (*Tenebrio molitor*) to increase average larval weight and total frass weight [12,13].

Despite the popularity of crickets as a feeder insect, there is minimal research related to biological feed additive use in any cricket species. Our research aims to assess the outcome of adding a commercial formulation of *S. cerevisiae* fermentation-based postbiotic to mass-reared *Acheta domesticus*. We hypothesize that crickets receiving feed enhanced with a *S. cerevisiae* fermentation-based postbiotic will produce a higher total biomass (increased weight per cricket and/or increased number of crickets), will have decreased viral loads, and will have significant differences between microbiome populations than those being fed the control cricket feed. The authors believe the addition of this postbiotic will improve cricket health, resulting in increased cricket yields and a more economical and environmentally friendly protein source for humans and animals.

## 2. Materials and Methods

### 2.1. Rearing Set Up and Feed Preparation

For the entirety of the experiment (32 days), crickets were raised in standard production bins (80 cm × 60.1 cm × 32 cm) at a commercial cricket operation (Fluker’s Cricket Farm, Inc., Port Allen, LA, USA). Bins were covered, vertically stacked, and outfitted with mesh ventilation covers on either end. Bins contained cardboard partitions to provide climbing and hiding structures. Crickets were provided with ad libitum feed and filtered tap water throughout the entirety of the experiment. The feed provided to the crickets was a commercial cricket chow, in mash form, with corn, soy, and wheat as the primary ingredients (Fluker Farms, Port Allen, LA, USA). Feed and water were provided by proprietary means that limited the need for workers to access the bins during the entire course of the experiment. Dead crickets and frass were not removed until the conclusion of the study. A total of 15 bins were used to test a commercial *S. cerevisiae* fermentation-based postbiotic (NaturSafe™, Diamond V, Cedar Rapids, IA, USA). The crickets were divided into three diet groups and fed for the entirety of the experiment: control cricket feed (Control, n = 5 bins), cricket feed supplemented with 0.25% postbiotic (P25, n = 5 bins), and cricket feed supplemented with 0.5% postbiotic (P5, n = 5 bins) on an as fed basis. The postbiotic powder was mixed with the cricket feed in a 55-gallon drum and rolled for at least one hour to ensure thorough mixing prior to distribution to the cricket bins. Replicates of this experiment were performed asynchronously due to space constraints within the rearing facility and the number of crickets that could be utilized for research purposes at one time without interruption to product demands. Six bins (n = 2 for each treatment) were collected in November and nine bins (n = 3 for each treatment) were collected in April. The nutritional analyses, viral qPCR, and microbiome data were only acquired for the second set of replicates due to the loss of samples from the first run.

### 2.2. Cricket Set Up

Day-old, farm-raised, *A. domesticus* were counted using an automated counter and software (Sciotex PerfectCount, Coleman Technologies, Inc., Newton Square, PA, USA; PerfectCount V3.3.1103) to ensure even distribution across bins. Twelve thousand crickets were placed in each bin on Day 1 of the experiment. All bins were housed in a single building with median temperature and humidity being 32.6 °C (interquartile [25–75%] range: 32.4–33 °C) and 61.4% RH (interquartile [25–75%] range: 54.7–66.1%), respectively. Selection of bins in the building was performed using a random number generator (random.org). The bins were all stacked at the same height to limit differences in temperature stratification within the building. Crickets were not disturbed for the length of the experiment. Thirty-two days was pre-determined as being the average amount of time needed to grow a majority of the crickets (>50%) to their final adult molt as evidenced by the presence of chirping and full-length wings and ovipositors.

### 2.3. Total Biomass, Average Cricket Weight, and Survivability

On Day 32, all crickets were removed from each bin and transported to Louisiana State University School of Veterinary Medicine (Baton Rouge, LA, USA) for further processing, cold storage, and analysis. All crickets were anesthetized with isoflurane (Fluriso, MWI Vetone, Boise, ID, USA)-soaked cotton balls until they ceased moving, and then frozen and stored at −80 °C to ensure a humane death [15]. The total weight of crickets from each bin was measured to determine total biomass (TBM). The average cricket weight was then determined by weighing a group of 100 randomly chosen crickets from each bin (random.org) and dividing by 100. Final counts were performed by hand to ensure accuracy and percent survival was calculated as the final count divided by the initial count multiplied by 100.

### 2.4. Nutritional Analyses

Two replicates of each diet (30 g each) were collected on Day 1 for nutritional analysis. On Day 32, two replicates of crickets from three bins in each treatment (30 g each, n = 6/treatment group) were removed for nutritional analysis. All samples were shipped on ice to Dairy One Forage Laboratory (Ithaca, NY, USA). Moisture, dry matter, crude protein, crude fat, nitrogen-free extract (NFE, on diet only), and ash were measured using testing methods developed for animal feed and pet foods. Insects were first homogenized and dried in a forced air oven at 105 °C (221 °F) for three hours to obtain dry matter content. Additional methods followed the AOAC handbook (Crude Protein-AOAC 992.15-Dumas combustion method, nitrogen-to-protein conversion factor = 6.25, Crude Fiber-AOAC approved procedure Ba 6a-05, Crude Fat-Acid Hydrolysis: AOAC 954.02, Ether Extraction: AOAC 2003.05, Ash-AOAC Method 942.05; Nitrogen-free extract calculated as %NFE = 100 − [%moisture + %CF + %CP + %EE + %Ash]) [16].

### 2.5. Cricket Dissections and DNA Extraction

Using a random number generator (random.org), five crickets were selected from three bins in each treatment group (n = 5 per bin, n = 15/treatment group). These crickets were sexed, weighed, and then sprayed in 70% ethanol and allowed to sit for five minutes to sterilize the exoskeleton prior to dissection. The entire gastrointestinal tract was removed from each individual and placed in a microcentrifuge tube in preparation for DNA extraction. DNA was extracted using a Qiagen DNeasy Blood and Tissue kit (Qiagen, Hilden, Germany) following the tissues and rodent tails protocol described in the manufacturer’s manual. The DNA concentration for each sample was measured and assessed for purity (A260/A280 ratio = 1.8–2.0) using a Nanodrop ND-1000 (Thermo Fisher Scientific, Wilmington, DE, USA). A blank control sample using only the extraction kit components was included to account for any bacteria that might have been present in the kit reagents.

### 2.6. Viral qPCR

Extracted DNA from the dissected GI tissue (n = 15 per treatment group) was tested for three viruses known to infect *A. domesticus*, AdDV, AdVVV, and CrIV. Quantitative real-time PCR assays were performed for AdDV, AdVVV, and CrIV using primers and probes that have been previously published [7,17]. The PCR reactions were run using TaqMan^®^ Universal PCR Master Mix and a QuantStudio 12K Flex Real-Time PCR System (Applied Biosystems, Foster City, CA, USA). Constructed plasmids that included the corresponding DNA sequences and nuclease-free water were used as positive and negative controls, respectively, for each virus. A standard curve was included for each virus that produced efficiencies between 90 and 95%. For AdDV and AdVVV, the reaction was set up to include 400 nM of forward and reverse primers each, 250 nM of probe, and 2 μL of DNA template in a 10 μL reaction. For CrIV, the reaction was set up to include 900 nM of forward and reverse primers each, 250 nM of probe, and 1 μL of DNA template in a 10 μL reaction. The cycling conditions for AdDV and AdVVV were as follows: (1) 98 °C for two minutes to activate the enzyme, and (2) 40 cycles at 98 °C for 10 seconds to allow for denaturation and 58 °C at 30 s each for annealing and extension. The cycling conditions for CrIV were as follows: (1) 50 °C for two minutes, and (2) 40 cycles at 95 °C for 15 seconds and 65 °C at 60 s each. All assays were run in triplicate. A CT value of ≥35.000 was used as the cutoff for consideration as a negative test.

### 2.7. Viral qPCR

Extracted DNA from the dissected GI tissue were sent to Novogene Corporation (Durham, NC, USA) for 16S and fungal 18S amplicon metagenomic sequencing and analysis. The four extractions with the highest purity DNA and concentrations out of the original five per bin were the ones sent for analysis (n = 4 per bin, n = 12/treatment group). PCR amplification of the V3–V4 hypervariable region of the 16S rRNA gene and the V4 region of the 18S rRNA gene was performed using 15 µL of Phusion^®^ High—Fidelity PCR Master Mix (New England Biolabs, Ipswich, MA, USA); 0.2 µM of forward and reverse primers (16S V3–V4: 341F-CCTAYGG GRBGCASCAG and 806R-GGACTACNNGGGTATCTAAT; 18S V4: 528F-GCGGTAAT TCCAGCT CCAA and 706R-AATCCRAGAATTTCACCTCT), and approximately 10 ng of template DNA. Thermal cycling consisted of initial denaturation at 98 °C for 1 min, followed by 30 cycles of denaturation at 98 °C for 10 s, annealing at 50 °C for 30 s, and elongation at 72 °C for 30 s, then a final step of 72 °C for 5 min. PCR products were then assessed for quality by mixing the same volume of 1× loading buffer (contained SYBR green) with the PCR products and performing electrophoresis on 2% agarose gel for detection. PCR products were then mixed in equidensity ratios and purified with the Universal DNA Purification Kit (TianGen, Beijing, China, Catalog #: DP214). Sequencing libraries were generated using NEB Next^®^ Ultra™ II FS DNA PCR-free Library Prep Kit (New England Biolabs, USA, Catalog #: E7430L) following the manufacturer’s recommendations, and indices were added. The library was checked with Qubit and real-time PCR for quantification and a bioanalyzer for size distribution detection. Quantified libraries were pooled and sequenced to a depth of 100,000 reads via the Illumina NovoSeq 6000 platform (San Diego, CA, USA).

### 2.8. Bioinformatics Analysis Pipeline

Paired-end reads were truncated of unique barcodes and primer sequences using Python (V3.6.13) and cutadapt (V3.3), then merged together using FLASH (V1.2.11). Raw tags were filtered for quality using fastp (Version 0.23.1). Clean tags were compared with the Silva database using UCHIME Algorithm to detect chimera sequences. Effective tags were obtained by removing the chimera sequences with the vsearch (V2.16.0) package. Sequence analysis was performed by Uparse (v7.0.1001). Sequences with ≥97% similarity were assigned to the same operational taxonomic unit (OTU). Reads produced in the blank extraction sample were removed from the final OTU table prior to bioinformatic analysis.

16S and 18S OTU tables were then uploaded to the Microbiome Analyst website (https://www.microbiomeanalyst.ca, accessed on 5 November 2024) along with accompanying taxonomy and metadata information [18]. OTUs were filtered out if at least 10% of the samples did not contain at least two reads each. Cumulative sum scaling was also performed to normalize the various library sizes between each sample. Rarefaction curves and alpha diversity metrics were performed on non-filtered data, while the remaining analyses were performed using the filtered data. Alpha diversity metrics performed include Observed Species, Chao1, Shannon, and Simpson using ANOVA as our base statistical method with post hoc t-tests as needed to determine significance between two levels of an independent variable. Beta diversity was performed using non-metric multi-dimensional scaling (NMDS) as the distance-based ordination technique, Bray–Curtis dissimilarity as the distance method, and PERMNOVA as the statistical method. Important taxonomic features between groups were identified with a combination of single-factor (using EdgeR as the statistical method), LEfSe (using a false discovery rate or FDR as a *p*-value adjustment of *p* = 0.1 and log linear discriminate analysis score of 2.0), and random forest analyses (using 500 trees, 7 predicters, and randomness setting set to “on”).

### 2.9. Statistical Analysis

The distributions of the continuous data were evaluated using the Shapiro–Wilk test, skewness, kurtosis, and q-q plots. Non-normal data were log-transformed; however, if the data could not be normalized then non-parametric methods were used to analyze those data. A one-way ANOVA was used to evaluate the normally distributed variables of TBM, percent survival, all nutritional data with the exception of crude fat in the crickets, and the qPCR CT values for AdDV and AdVVV. A Kruskal–Wallis test was performed for average cricket body weight, starting counts, crude fat per cricket, and CrIV CT values. Dunnett t-tests or Mann–Whitney U tests were used as the method for post hoc testing to look for differences between the groups when appropriate. A 3 × 2 Chi Square was used to assess viral prevalence between groups when prevalence was less than 100%. A *p* ≤ 0.05 was used to determine significance for all tests; however, given our small sample sizes and the high risk of a Type II error occurring, 0.5 < *p* < 0.10 was used to indicate a trend towards significance. For *p* values within this range, post hoc testing was utilized when appropriate to help determine where differences lied between treatment groups [19,20,21]. SPSS 28.0 (IBM Statistics, Armonk, NY, USA) was used to analyze the data.

## 3. Results

### 3.1. Total Biomass, Average Cricket Weight, and Survivability

All variables were normally distributed, except for average body weight per cricket and starting counts. There was no difference in the starting counts for all bins (KW(2) = 1.159, *p* = 0.560). The TBM (g), average weight per cricket (mg), and the percent survivability on Day 32 for each treatment are listed in Table 1. There was a trend towards significance between the three diet groups when analyzed with a one-way ANOVA for TBM (F(2,12) = 3.823, *p* = 0.052) and for percent survival (F(2,12) = 3.667, *p* = 0.057); there was no significant difference in average weight per cricket (KW(2) = 1.523, *p* = 0.470). Post hoc testing revealed that crickets offered the 0.5% supplemented diet were found to have a significantly higher TBM and percent survival than either the 0.25% or control diet.

### 3.2. Nutritional Analysis

All variables were normally distributed except for crude fat in the cricket samples (see Table 2). No significant differences were found between the three diet formulations or between the three treatment groups for any of the nutritional analyses except for ash content between cricket samples (feed-dry matter: F(2,3) = 1.762, *p* = 0.312; crude protein: F(2,3) = 0.022, *p* = 0.978; NFE: F(2,3) = 0.163, *p* = 0.857; crude fat: F(2,3) = 0.148, *p* = 0.868; fiber: F(2,3) = 0.396, *p* = 0.704; ash: F(2,3) = 1.067, *p* = 0.447; crickets-dry matter: F(2,15) = 2.581, *p* = 0.109; crude protein: F(2,15) = 1.004, *p* = 0.390; fiber: F(2,15) = 2.791, *p* = 0.093; crude fat: KW(2) = 0.012, *p* = 0.994; ash: F(2,15) = 3.514, *p* = 0.056).

### 3.3. Viral qPCR

There were no significant differences in viral prevalence or load between diet treatments for any of the three viruses tested. The prevalence of CrIV was 100% in all samples, with no differences in CT values between groups (KW(2) = 0.613, *p* = 0.736). AdDV had a prevalence of 26.7% (4/15) in both the control and the 0.25% diet group and a 6.7% prevalence (1/15) in the 0.5% diet group (χ^2^(2) = 2.5, *p* = 0.287). Of those samples that were positive, there was no difference in Ct values between groups (F(2,6) = 0.447, *p* = 0.659). The prevalences for AdVVV were 73.3% (11/15), 80% (12/15), and 60% (9/15) in the control, 0.25%, and 0.5% diets, respectively, (χ^2^(2) = 1.514, *p* = 0.47). Of those samples that were positive, there was no difference in CT values between groups (F(2,29) = 0.752, *p* = 0.481).

### 3.4. 16S/18S Microbiome Analysis

16S amplicon sequencing produced 6,583,957 raw paired-end reads with 3,361,691 total reads classified to an OTU. 18S amplicon sequencing produced 4,958,269 raw paired-end reads with 3,032,383 total reads classified to an OTU. A total of 295 OTUs for 16S sequencing and 222 OTUs for 18S were established and uploaded to the Microbiome Analyst website. Low-abundance data filtering removed 101 and 99 taxonomic features from the 16S and 18S data, respectively. Rarefaction curves confirmed ample sequencing depth for the experiment (Appendix A).

Figure 1 and Figure 2 are bar graphs displaying the relative abundance of different classes between groups for both 16S and 18S amplicon data. For the 16S data, we see large contributions from the *Bacteroidia* class in all three diet groups. There is also an increase in the *Clostridia* class of bacteria as the postbiotic inclusion percentage increases (control 95% CI = 20.8 ± 11.2%; P25 95% CI = 27.8 ± 8.8%; P5 95% CI = 40.5 ± 9.9%). *Gamma-proteobacteria* are also present in high numbers in some of the samples from the control group and the 0.25% group; however, there is very little contribution from *Gamma-proteobacteria* in the 0.5% group (control 95% CI = 6.9 ± 7.8%; P25 95% CI = 5.7 ± 7.3%; P5 95% CI = 0.7 ± 0.5%). For the 18S data, we see large contributions from classes associated with plant material, *Liliopsida* (wheat and other close relatives) and unidentified *Streptophyta* (soybean and other close relatives). In the 0.5% diet group, we see a marked increase in the *Conoidasida* class of Apicomplexan protists (control 95% CI = 8.6 ± 12.6%; P25 95% CI = 8.3 ± 13.3%; P5 95% CI = 23.9 ± 13%). There are highly variable percentages of *Saccharomycetes* in all groups (control 95% CI = 8.2 ± 11.5%; P25 95% CI = 14.9 ± 17.8%; P5 95% CI = 12.1 ± 15.8%). Figure 3 is a Venn diagram showing the number of OTUs shared between each of the groups and the number of unique OTUs per group.

Alpha diversity metrics are used to show species richness and evenness between groups. Observed Species and Chao1 are metrics used to assess species richness with higher values indicating more species being present in a population. In the 16S data (Figure 4), there was a trend towards significance in the observed species metric (F(2,9) = 3.177, *p* = 0.055), with a specific difference between the control diet and 0.5% diet groups (t = −2.192, *p* = 0.040); differences between the control and 0.25% diet (t = 1.820, *p* = 0.085) and 0.25% and 0.5% diets (t = 0.998, *p* = 0.333) were not significant. There was also a trend towards significance in the Chao1 metric (F(2,9) = 3.053, *p* = 0.061), with differences between the control and 0.25% diet (t = 2.250, *p* = 0.035) and the control and 0.5% diet (t = 2.201, *p* = 0.039). The difference between 0.25% and 0.5% was non-significant (t = 0.099, *p* = 0.922). Shannon and Simpson metrics are more commonly used to assess the evenness of the species distribution with higher values being given to populations with more dominant species present and fewer rare species present. Neither of these metrics were significant for the 16S data (Shannon: F(2,9) = 1.506, *p* = 0.237; Simpson: F(2,9) = 1.957, *p* = 0.157), indicating that all three groups were similar in evenness scores.

In the 18S data (Figure 5), all four diversity metrics lacked significance between groups (Observed Species: F(2,9) = 0.044, *p* = 0.957; Chao1: F(2,9) = 0.150, *p* = 0.861; Shannon: F(2,9) = 0.524, *p* = 0.597; Simpson: F(2,9) = 0.577, *p* = 0.567). This indicates similar amounts of species richness and evenness in each of the three groups.

Beta diversity is the measure of similarity/dissimilarity between populations. There were no significant differences between any group for either the 16S (F(2,9) = 1.390, *p* = 0.112, r^2^ = 0.078, NMDS stress = 0.155) or 18S data (F(2,9) = 0.685, *p* = 0.700, r^2^ = 0.040, NMDS stress = 0.199) (Figure 6). The stress level for each plot is between 0.1 and 0.2, indicating a fair but not excellent representation of how the data are displayed in a 2D model.

Single-factor analysis identified 15 significant features at the OTU level for 16S amplicon data. Crickets receiving the yeast postbiotics saw increased levels of several OTUs including the genera *Akkermansia*, *Catenibacillus*, and *Odoribacter*; the families *Beijerinckiaceae*, *Desulfovibrionaceae*, [*Eubacterium*] *coprostanoligenes* group, *Oscillospiraceae*, and *Peptococcocaeae*; the order *Peptostreptococcales-Tissierellales*; and the class *Bacteroidia* (Appendix A). LEfSe analysis revealed no significant taxonomic features using the cut-off criteria specified previously. However, *Lachnospiraceae, Peptococcaceae, Peptostreptococcales-Tissierellales,* and *Odoribacter* spp. showed the greatest divergence between diet groups. These features, along with their LDA scores, can be found in Appendix A. The top 10 most important features from the random forest analysis are presented below in Appendix A.

Single-factor analysis for the 18S amplicon data identified seven significant features. The genus *Saccharomyces* was significantly increased in the two diet groups receiving the *S. cerevisiae* postbiotic. The genera *Sistotrema* and *Thalassiosira* and the species *Colletotrichium truncatum* were significantly increased in the control diet group (Appendix A). The remaining significant features identified were plant material or cricket DNA that varied between groups and are unlikely to be of any clinical significance. LEfse analysis again revealed no significant features between the three groups with the previously described settings. However, *Malassezia restricta*, *Leidyana erratica*, *Saccharomyces* spp., and *Aspergillaceae* were the features that showed the greatest divergence between diet groups. These features and their LDA scores are presented in Appendix A. The top 10 most important features identified with random forest analysis are presented in Appendix A.

## 4. Discussion

The primary goal of this study was to measure the impact of adding a commercial *S. cerevisiae* postbiotic product to the diet of *A. domesticus* under mass rearing conditions. Based on previous studies performed in other mass-reared insect species [13], the authors hypothesized that the addition of a postbiotic would likely result in increased cricket weights and decreased mortality, all of which would increase the total biomass of crickets produced. Supplementing the diet with *S. cerevisiae* at 0.5% did not result in significantly higher cricket weights, but there was a trend towards significance with an increase in TBM, mostly due to increased survivability. Due to the small sample size (which was limited due to production constraints on an active insect farm), the authors are highly certain that a type II error is associated with the *p* values being >0.05 (*p* = 0.052 for TBM and *p* = 0.057 for percent survival). Additionally, the experimental set up of having three treatment groups rather than two removed the authors’ ability to perform unidirectional hypothesis testing. As a result, the authors decided to include post hoc testing on *p* values < 0.10 instead of just those that fell < 0.05. While the 0.25% yeast inclusion diet also produced bins with a higher total biomass as compared to the control diet, these gains were not found to be significant, and this inclusion level also produced bins with the lowest survivability. It is possible that one or more of the 0.25% bins were affected by an outside factor that increased mortality in those bins and hence lowered the average total biomass. However, due to our study design with a small sample size and low replicate counts, we were unable to overcome the variable results that may have occurred within bins. The study was designed to mimic mass rearing conditions and was performed at a commercial insect rearing facility which limited our ability to have more than three replicates of each group at any one time. Additional studies could be performed to truly determine the best yeast inclusion percentage to add to the diet (and to determine the best benefit-to-cost ratio). This pilot experiment allowed proof of concept under industrial conditions while producing favorable results, a feature that has so far been lacking within the literature for most feed additive studies involving insects.

The average weight per cricket increased 2.9% and 8.4% with the 0.25% and 0.5% supplemented diets, respectively, as compared to the control diet (% increase = (treatment avg body weights − control avg body weight)/control avg body weight); however, this finding was not significant. Another study [22] looking at the addition of *S. cerevisiae* in *Gryllus bimaculatus* (two-spotted crickets) diets did find a significant increase in weights between crickets being fed a standard control diet and those with live *S. cerevisiae* added (47.3% increase). The difference in weight gain between our study and Yang et al. (2023) [22] could be associated with the use of a different cricket species, unequal sex ratios, a different yeast formulation and concentration (commercial postbiotic product with other bioactive compounds present vs. a live single species culture), or the underlying disease status of the crickets. It is important to consider these variables when comparing results because cricket species can differ in body weight, sex biased studies can result in higher biomasses for females or lower biomasses for males, and differences in growth can be impacted by the organs impacted by different pathogens.

The other noticeable difference between our study and Yang et al.’s (2023) [22] was that they found no differences in mortality rates between the control and experimental diets (54.6–61.7% mortality rate), whereas we found a significant decrease in mortality for those receiving the 0.5% yeast inclusion (69.1% mortality rate) as compared to that of the control diet (74.8% mortality rate). The present study’s mortality rates are quite high compared to other published rates for lab-reared *A. domesticus* (3–55.5% mortality) [23,24,25,26,27,28]. Historic and current mortality rates on this farm are typically more in line with the published rates; however, fluctuations do periodically occur (unpublished data). In general, mortality rates for lab-reared and farm-raised *A. domesticus* are usually high. This could be due to a number of factors including inadequate environmental factors such as temperature or humidity, high rearing densities, or infectious causes such as bacterial, viral, fungal, parasitic, or protozoal diseases. For the *A. domesticus* in our study, viruses were a common finding. Two of the three viruses we tested for are known to cause high morbidity and mortality in *A. domesticus* populations [6,29,30,31], with the third virus (AdVVV) still having unknown effects. Therefore, the authors hypothesize that the inclusion of the postbiotic to our study population, while not capable of clearing the virus or decreasing the viral load, may have supported the immune system (theoretically through the changes seen in the microbiome) of a stressed population and reduced the mortality rate to a lower level. The Yang et al. study [22] did not look into the disease status of their population and *Gryllus* and *Gryllodes* crickets have been shown to be less susceptible to the effects of these viruses, which may be why they did not see any change in mortality rates. Future cost–benefit analyses should be conducted to determine if the addition of nutritional supplements like pre-, pro-, and post-biotics can offset the high losses often seen in production settings.

Although multiple studies exist for examining the use of biological feed additives in other mass-reared insect species [13], the Yang et al. (2023) [22] study is the only publication that has evaluated the use of a probiotic on any cricket species. One other study was found that looked at the growth performance of *A. domesticus* and *G. bimaculatus* when fed on agro-byproducts including one diet that included spent yeast [32]. Both species of crickets in this study had lower weight gain than the other diets used. The authors of that study hypothesized that the higher protein content of the spent yeast (49.7% DM) may explain the slower growth due to crickets needing to expend more energy in the excretion of excess protein in the forms of uric acid and ammonia. This is a common finding in insect diet trials, where herbivorous and omnivorous insects grow faster or live longer with a higher carbohydrate-to-protein ratio [32,33].

Other studies specifically using *S. cerevisiae* or non-specified brewer’s or baker’s yeast exist for several species of mealworms (*Tenebrio molitor*, *Zophobas morio*, and *Alphitobius diaperinus),* silkworms (*B. mori*), black soldier fly (BSF) larvae (*H. illucens*), and fruit flies (*Drosophila suzukii*) [12,13,33,34,35,36,37,38,39,40,41]. These studies cover multiple different study parameters from growth, development times, mortality, feed conversion rates, nutrient composition, and silk production. Some had positive effects on the insects [12,33,34,36,40,41] while others had neutral to negative effects on the insects [35,37,38,39]. Overall, *S. cerevisiae* seems to lead to positive effects in insects (decreased mortality, shorter development times, increased larval weights) more often than it has a negative effect (higher larval mortality). The variable results seen from each of these studies highlight the importance of how much yeast product is added to the control diet and what formulation is being used (live cells versus yeast product and metabolites). It is also important to note whether the yeast is capable of inhabiting and reproducing in the gut or if it is acting simply as a source of nutrients and metabolites because this could have significant results on the nutritional profile, growth and development, microbiome, and immune system. In our case, the commercial product used did not include live yeast cells but rather post-fermentation *S. cerevisiae* nutrients and metabolites (such as B vitamins, amino acids, and volatile fatty acids), and hence its classification as a postbiotic. In vertebrates, *S. cerevisiae* postbiotics have been shown to reduce gut inflammation, stimulate the immune system, support mucosal defense, and increase concentrations of short chain fatty acids [42]. Not all these benefits have been identified in invertebrate animals, but there is a good reason to believe that *S. cerevisiae* postbiotics can confer several benefits to insects as well.

In addition to wanting to speed up development times and increasing TBM for production reasons, one of the most pressing reasons for experimenting with biological feed additives in mass-reared insects is because of the high morbidity and mortality rates seen within the insect industry [3,29]. Disease transmission is common in mass-reared insects with viral diseases being the most implicated for losses in cricket production [6,43]. Other recent work from the authors suggests that bacterial dysbiosis may also be a concern [44] and the use of antimicrobials in this newly emerging field of insect agriculture would be less than ideal given the rise in antimicrobial resistance in other areas of the food industry. With this in mind, the authors analyzed each cricket sample for the presence of three different viruses known to infect *Acheta domesticus*, AdDV, AdVVV, and CrIV. While our results did not show any conclusive reduction in these viruses, there was a trend towards less viral prevalence with increasing amounts of yeast product in the diet; however, a much larger sample size (calculated n = 126 vs. the current n = 45) would be needed to determine if yeast inclusion leads to lower viral prevalence. Higher yeast inclusion rates and different types of feed additives could also be experimented with to see if those changes would lead to more beneficial immunomodulatory effects.

In regard to the effect on the microbiome, we looked for changes that may have occurred within the 16S (bacterial) and 18S (eukaryotic) microbial profiles because these changes may have an impact on the crickets’ overall health. Similarly to other cricket 16S microbiome studies, the most dominant phyla were *Bacteroidetes* (*Bacteroidota*), *Firmicutes* (*Bacillota*), and *Proteobacteria* (*Pseudomonadota*). Further classification revealed high percentages of *Bacteroida*, *Clostridia*, and *Bacilli* in all samples. Using 95% confidence intervals at the class level of bacteria, we were unable to determine any significant differences between the three diet groups. For 18S data, the most dominant phyla were *Streptophyta* (plant material from diet), *Apicomplexa* (protists), and *Ascomycota* (fungi). Further classification revealed high percentages of *Poales* and *Fabales* (order levels for wheat, corn, soy, and other close relatives), *Conoidasida* (protists), and *Saccharomycetes*. Although there is strong visual evidence to suggest significance between diet groups for these classifications, the constructed 95% confidence intervals determined that these differences were not significant between diet groups. Further analysis methods more suited to metagenomic data such as alpha and beta diversity and differential abundance analyses would therefore be needed to look for differences.

For alpha diversity, the 16S data did reveal slight differences between the yeast and control diets when analyzing for Observed Species. This metric is used to measure for species richness between samples and our results would indicate that the addition of a yeast product to the diet slightly increased the number of bacterial species present, which is usually an indicator of better gut health [45]. 18S amplicon data did not show any differences between the richness of eukaryotic organisms. On the other hand, beta diversity is a measure of similarity/dissimilarity between populations, and there were no significant differences for either the 16S or 18S data; however, the NMDS plot for the 16S data did show a tighter clustering of the yeast samples compared to the control diet. While the community composition is similar between groups, the yeast samples were more similar to each other than they are to some of the control diet samples.

Single-factor, LEfSe, and random forest analyses were used to determine if any specific OTUs differed between groups. For those OTUs that were increased in the yeast-inclusive groups, many of them are considered healthy gut commensal organisms with low to no pathogenicity reported in any species. For example, *Akkermansia* and *Odoribacter* spp. are known to be associated with lower gut inflammation, and low to absent numbers of these genera are often associated with pro-inflammatory diseases in humans. These bacteria are known to produce short-chain fatty acids that are beneficial to the host [46]. *Catenibacillus* is another known producer of short-chain fatty acids and has been found to decrease levels of pro-inflammatory cytokines [47]. *Bacteroides* (and by extension *Bacteroidia*) and *Eubacterium* spp. are two of the most common genera found in the human gut and have been found to be very important for suppressing inflammation, immunomodulation, and maintaining homeostasis [48,49]. Previous microbiome studies have shown that these two genera are also prevalent in healthy cricket guts and are likely important organisms [44]. In Yang et al. (2023) [22], researchers also saw a significant increase at the *Bacteroidetes* phylum level for crickets receiving the *S. cerevisiae* probiotic, matching our findings. This can be compared to the finding of decreased percentages of *Bacteroidetes* and *Firmicutes* in sick versus healthy crickets [44]. *Peptococcoceae* and *Desulfovibrionaceae* are common bacterial families found in both human and termite guts. In termites, *Desulfovibrionaceae* was highest in wild individuals and lab-reared individuals being fed low-protein diets. These bacterial species are important for sulfate reduction, recycling of nutrients, and could play a role in acetate formation, all of which are important for gut health [50]. For those bacterial OTUs that were most abundant in the control group, only the genus *Robinsoniella* is potentially concerning. The remainder are normal gut microbes that are usually associated with positive effects. However, the genus *Robinsoniella* only has one identified species, *Robinsoniella peoriensis*, and while there is not much information related to this species in insects, there are several recent articles in human medicine discussing it as an emerging pathogen capable of causing serious infections including bacteremia, osteomyelitis, and pyometra [49]. This genus was also identified in the Yang et al. study (2023) [22] in *Gryllodes* crickets receiving the control diet. Additional research should be performed on this bacterial species to determine if there is indeed pathogenicity associated with its presence in insects.

For the 18S amplicon data, OTUs that were increased in the yeast-inclusive diets included *Leidyana erratica*, *Saccharomyces*, and a few plant-related organisms. An elevation of *Saccharomyces* was an expected outcome given that it was supplemented in the diet even if the product used did not contain live cells. The control group still produced some reads of *Saccharomyces* despite the lack of supplementation. This may be an indication that *Saccharomyces* may already be a part of the normal mycobiome of the *A. domesticus* cricket. To the authors’ knowledge, this is the first study to evaluate 18S amplicon data for this cricket species.

The increase in *Leidyana erratica*, an Apicomplexan, on the other hand, was an unexpected finding. This species of gregarine protist has been identified in *Gryllus* crickets [51,52], but to the authors’ knowledge this is the first report of it potentially being found in *A. domesticus*. Isolation of the organism and morphological confirmation or full sequencing would be necessary to truly confirm its presence within a new species of cricket. Apicomplexans are typically considered obligate parasites, with most gregarines (exception being *Cryptosporidium* spp.) only being found in invertebrate species [53]. However, there is much debate as to whether gregarine species are actually parasites because most infections are considered benign. There is abundant research showing positive, neutral, and negative effects to their hosts depending on the species [53,54]. In *T. molitor*, two studies have shown that *Gregarina* spp. were responsible for enlarged host growth and positive impacts on host development, fitness, and longevity [55,56]. However, other studies have demonstrated their destruction to host gut cells and decreased longevity in individuals with heavy infectious loads [57,58,59]. In field crickets (*Gryllus* spp.), one study found that gregarines had no effect on weight, longevity, or fecundity when fed ad libitum, but when fed suboptimal diets, the infected crickets suffered from more severe weight loss and reduced longevity as compared to control crickets [60]. Similarly to opportunistic gut bacteria, there appears to be a spectrum where under certain conditions the bacterium is beneficial to the gut, but under others, including heavy overgrowth and in times of stress, it can become a fatal pathogen to the host. As for this study, the increased presence of *L. erratica* in the crickets receiving the 0.5% diet appears to correspond with increased cricket survival. However, more research into this particular relationship is needed to determine whether *L. erratica* is acting as a beneficial endosymbiont or if it is only present as a benign passenger within the gut.

OTUs that were most abundant in the control group included *Colletotrichium truncatum*, *Sistotrema*, *Thalassiosira*, and the family *Aspergillaceae*. There is very little to no literature published regarding *Colletotrichium truncatum* (a plant fungal pathogen), *Sistotrema* (a fungus commonly found in human breastmilk and the human gut), and *Thalassiosira* (a marine diatom) within any insect gut microbiomes. Therefore, their clinical significance in control cricket guts is currently unknown. More research would be needed to determine if these organisms hold any kind of pathogenic threat to insects or if they are just benign inhabitants or transient passengers. However, members of the *Aspergillaceae* family are commonly known to be opportunistic pathogens to several different species, including various insects. *Aspergillus* spp. have been found to be a naturally occurring entomopathogenic fungus to certain moth larvae [61] and have been found to be an effective larvicide against certain species of mosquito larvae [62]. Three different species of *Aspergillus* (*A. flavus*, *A. niger*, and *A. phoenicis*) have been found to be the causative agents of “white fungus” and stonebrood in honeybee larvae and adults [63]. More beneficial effects from gut colonization are possible, as it has been found in many different species of insects including honeybees, termites, locusts, palm weevils, and beetles [62,64,65]. In honeybees, direct evidence of a mutualistic symbiosis is lacking, but *Aspergillus* spp. could play a role in competition with other more pathogenic *Aspergillus* strains, inhibit the growth of other pathogens and parasites, assist with detoxification, or help with the production of enzymes, nutrients, and antibacterial substances [63]. Whether it provides any benefits to cricket species remains to be investigated.

This study provided new information to the field of insects for feed and food. Not only is it one of the first studies to investigate a postbiotic product in a cricket species, but it is also the first to provide 18S amplicon data for any cricket species. The results of this study demonstrate that much more work needs to be performed to expand our knowledge of the cricket’s total intestinal microbiome (bacterial, fungal, viral, and protozoal) and determining what kind of roles fungi and protists play. Arming ourselves with more knowledge on how the cricket and its associated microbes work together (or against each other) is important for us to determine how to best implement changes to diet, husbandry, and disease management within the commercial cricket rearing industry.

## 5. Conclusions

Although further work remains to be performed, this study was able to provide evidence that a commercial *S. cerevisiae* fermentation-based postbiotic (NaturSafe™) added at an inclusion rate of 0.5% as fed could be beneficial to increasing total biomass and reducing mortality in crickets grown under commercial rearing conditions. Alterations to the 16S microbiome in the way of increased abundances of presumed beneficial bacteria (*Akkermansia, Catenibacillus*, and *Odoribacter*) are likely linked to the increased production values rather than reductions in harmful bacteria or viral agents. Alterations to the 18S microbiome were minimal and will require more study to determine their significance. Additional work is needed to determine the best effective dose and whether a live *S. cerevisiae* product produces similar or better results than the one used in this study.

## Figures and Tables

**Figure 1 insects-16-00702-f001:**
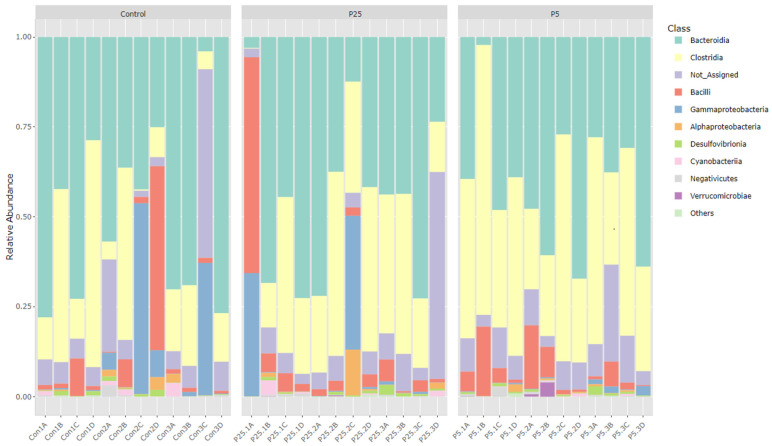
Relative abundance of 16S classes across treatment groups.

**Figure 2 insects-16-00702-f002:**
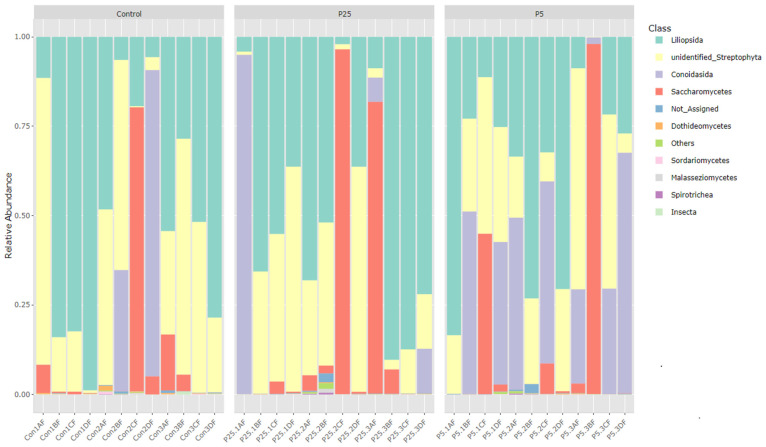
Relative abundance of 18S classes across treatment groups.

**Figure 3 insects-16-00702-f003:**
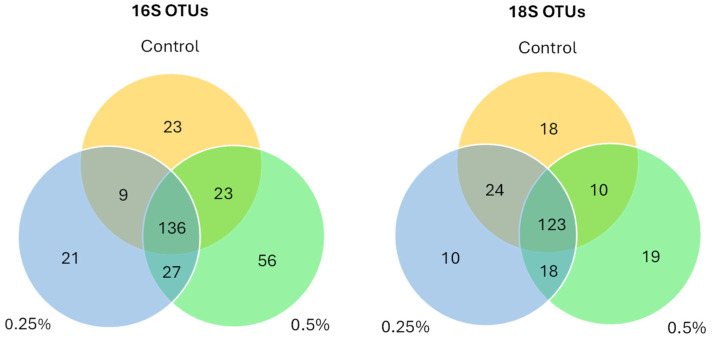
Venn diagrams showing shared and unique OTUs across treatment groups.

**Figure 4 insects-16-00702-f004:**
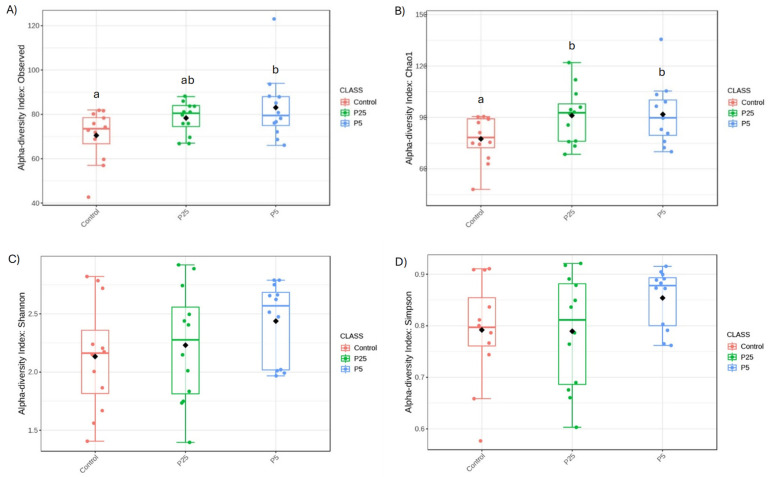
Alpha diversity metrics for 16S amplicon data. Panels A and B show species richness/abundance between groups, (**A**) Observed species, (**B**) Chao1. Panels (**C**) and (**D**) show species evenness between groups., (**C**) Shannon, (**D**) Simpson. P25 and P5 labels indicate 0.25 and 0.5% groups, respectively. ^a,b^—Different superscript letters indicate significant differences (*p* ≤ 0.05) between groups with post hoc testing.

**Figure 5 insects-16-00702-f005:**
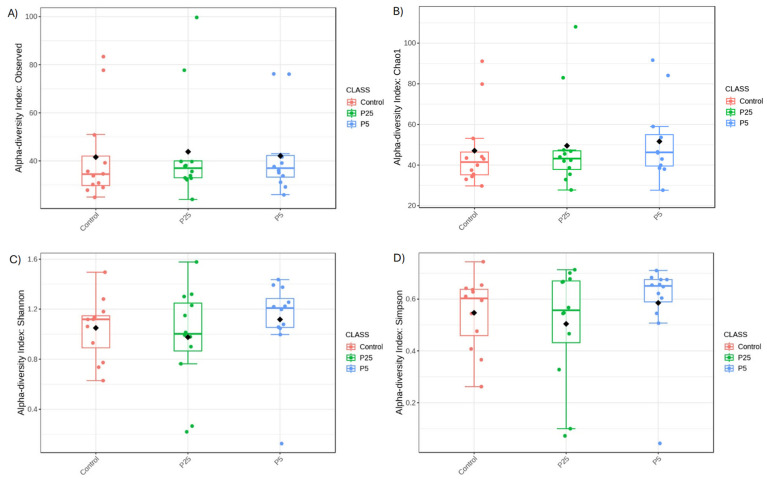
Alpha diversity metrics for 18S amplicon data. Panels A and B show species richness/abundance between groups, (**A**) Observed species, (**B**) Chao1. Panels (**C**,**D**) show species evenness between groups, (**C**) Shannon, (**D**) Simpson. There were no significant differences seen between any group with any of the four measured indices. P25 and P5 labels indicate 0.25 and 0.5% groups, respectively.

**Figure 6 insects-16-00702-f006:**
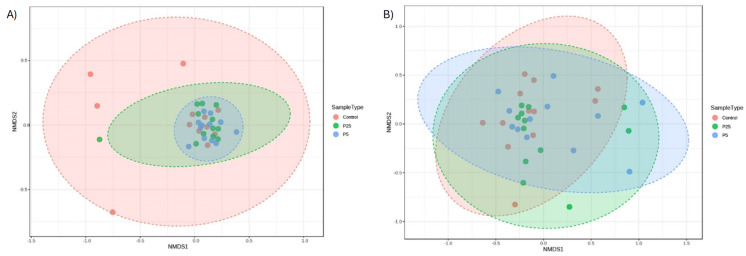
Beta diversity plots for 16S amplicon data and 18S amplicon data. (**A**) 16S data displayed using an NMDS plot. Treatment groups are tightly clustered with larger variance present in the control group. (**B**) 18S data displayed using an NMDS plot. Samples from all groups are heavily overlapped, indicating similarity. P25 and P5 labels indicate 0.25 and 0.5% groups, respectively.

**Table 1 insects-16-00702-t001:** Total biomass, average weight, and percent survival of crickets receiving a diet supplemented with a *S. cerevisiae* postbiotic.

Treatment	Total Biomass (g) ^2^	Average Body Weight per Cricket (mg)	% Survival ^2^
Control (n = 5)	982 ± 165 ^a^	310 ± 27	25.33 ± 3.39 ^a^
0.25% (n = 5)	1005 ± 192 ^a^	319 ± 43	25.23 ± 3.75 ^a^
0.5% (n = 5)	1278 ± 206 ^b^	336 ± 43	30.91 ± 4.08 ^b^

Note: All values are presented as mean ± SD. ^2^ Differing superscripts indicate significant differences (*p* ≤ 0.05) between diet groups with post hoc testing.

**Table 2 insects-16-00702-t002:** Nutritional analysis of three experimental diets and cricket treatment groups supplemented with various concentrations of a *S. cerevisiae* postbiotic.

Substrate Analyzed	Dry Matter (%)	Crude Protein (% DM)	Fiber (% DM)	NFE ^1^ (% DM)	Crude Fat ^2^ (% DM)	Ash ^3^ (% DM)
** Feed **						
Control diet (n = 2)	89.1 ± 0.1	21.5 ± 0.4	6.9 ± 0.1	64.6 ± 0.4	5.6 ± 0.1	7.1 ± 0.1
0.25% diet (n = 2)	89.0 ± 0.1	21.5 ± 0.9	6.7 ± 0.5	65.0 ± 1.1	5.5 ± 0.4	6.9 ± 0.2
0.5% diet (n = 2)	89.3 ± 0.3	21.6 ± 0.0	6.7 ± 0.0	64.8 ± 0.2	5.6 ± 0.1	7.0 ± 0.1
** Crickets **						
Control (n = 6)	27.1 ± 0.4	65.5 ± 0.4	6.1 ± 0.2	N/A	22.2 (21.8–23.0)	6.4 ± 1.0 ^a^
0.25% treatment (n = 6)	27.4 ± 0.6	66.6 ± 1.0	6.6 ± 0.6	N/A	22.4 (19.9–22.7)	7.4 ± 0.4 ^b^
0.5% treatment (n = 6)	27.8 ± 0.5	65.7 ± 2.1	6.0 ± 0.6	N/A	22.3 (17.6–23.3)	6.8 ± 0.3 ^a,b^

Note: All values are presented as mean ± SD with the exception of crude fat for crickets. The values for this test were non-normally distributed and are presented as median and interquartile range (25–75%) in parentheses. ^1^ Nitrogen-free extract (a calculated measure for carbohydrates) was not performed on the cricket samples. ^2^ Crude fat for diet was performed by ether extraction (EE). Crude fat for crickets was performed by acid hydrolysis (AH). ^3^ Differing superscripts indicate significant differences (*p* ≤ 0.05) between diet groups with post hoc testing.

## Data Availability

The raw data supporting the conclusions of this article will be made available by the authors on request.

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
