# Peer review of "Determining the Effectiveness of *Saccharomyces cerevisiae* as a Postbiotic in Mass-Reared *Acheta domesticus* (House Cricket)"

_insects, 2025, doi:10.3390/insects16070702_

Round 1

Reviewer 1 Report

Comments and Suggestions for Authors

Dear Authors,

The manuscript presents an interesting research idea. However, several sections require revision, particularly the addition of more results and substantial changes to the discussion. Please refer to the enclosed file for detailed comments.

Best regards,

The reviewer

Author Response

Thank you for taking the time to review our manuscript. Several changes have been made according to your suggestions. We appreciate your thoroughness in your review. A line-by-line breakdown is attached.

Reviewer 2 Report

Comments and Suggestions for Authors

Thank you for writing this manuscript entitled “Determining the effectiveness of Saccharomyces cerevisiae as a postbiotic in mass reared Acheta domesticus (house cricket)” providing nice results on effects of postbiotic provision on house cricket microbiota and growth. The manuscript is clear and complete and the experimental design well structure.

I do have few concerns about the surface sterilization method as you dipped the insects in 70% EtOH for 5 minutes before dissecting them and in this way the reagent could had get into the insects by changing the microbial composition. Why did you choose for this method?

Kind regards

Author Response

Reviewer #2 comments: I do have few concerns about the surface sterilization method as you dipped the insects in 70% EtOH for 5 minutes before dissecting them and in this way the reagent could had get into the insects by changing the microbial composition. Why did you choose for this method?

Reply to reviewer 2: Thank you for taking the time to review our paper and we appreciate your comments. It is possible that the EtOH could have altered the internal microbial composition to a small degree, but the authors believe this methodology was necessary to ensure no contaminants were picked up from the surface of the cricket while cutting through the cuticle to get to the GI tract.  Routes of entry would have been through the tracheal system (which should be minor and not a huge source of concern for affecting the inner GI contents) or thorough the mouth or anus. The 70% EtOH was heavily sprayed on the crickets and allowed to sit for 5 minutes rather than full immersion in EtOH for the entire 5 minutes. So even if a little alcohol made it into the mouth or anus, we do not believe enough would have entered with this technique to massively change our microbiota composition. It is also difficult to remove the most orad section of the esophagus and the most distal section of the anus while dissecting so the sections most at risk of compromise would not have been heavily represented in our samples anyways. The authors decided to use this methodology based off of other previously published works. If you think a statement is needed within the paper to discuss this methodology and its possible affects on the microbiota content, please let us know. Thank you.

We have updated the text to say "sprayed" rather than immersed.

Round 2

Reviewer 1 Report

Comments and Suggestions for Authors

Dear Authors,

There are still some sections of the revised manuscript that require further correction. Please see the details in the attached file.

Best regards,

Reviewer

Author Response

Thank you taking the time to make additional comments on our manuscript. Please see the file below with our line by line responses/edits.
